# Clinical characteristics and severity prediction score of Adenovirus pneumonia in immunocompetent adult

Chao Hu[1☯], Ying Zeng[2☯], Zhi Zhong[2], Li Yang[2], Hui Li[1], Huan Ming Zhang[2], Hong Xia[3‡*], Ming Yan Jiang[1‡*]

1 Department of Respiratory and Critical Medicine, Xiangtan Central Hospital, Xiangtan, Hunan, People's Republic of China, 2 Department of Radiology, Xiangtan Central Hospital, Xiangtan, Hunan, People's Republic of China, 3 Department of Orthopedics, Xiangtan Central Hospital, Xiangtan, Hunan, People's Republic of China

☯ These authors contributed equally to this work.
‡ HX and MYJ also contributed equally to this work.
* jiangmingyan1979@163.com (MYJ); xtxiayidao@163.com (HX)

**Data Availability Statement:** All relevant data are within the paper and its Supporting Information files.

## Abstract

### Background

Compared with children and immunocompromised patients, Adenovirus pneumonia in immunocompetent adults is less common. Evaluation of the applicability of severity score in predicting intensive care unit (ICU) admission of Adenovirus pneumonia is limited.

### Methods

We retrospectively reviewed 50 Adenovirus pneumonia inpatients in Xiangtan Central Hospital from 2018 to 2020. Hospitalized patients with no pneumonia or immunosuppression were excluded. Clinical characteristics and chest image at the admission of all patients were collected. Severity scores, including Pneumonia severity index (PSI), CURB-65, SMART-COP, and PaO2/FiO2 combined lymphocyte were evaluated to compare the performance of ICU admission.

### Results

Fifty inpatients with Adenovirus pneumonia were selected, 27 (54%) non-ICU and 23 (46%) ICU. Most patients were men (40 [80.00%]). Age median was 46.0 (IQR 31.0–56.0). Patients who required ICU care (n = 23) were more likely to report dyspnea (13[56.52%] vs 6 [22.22%]; P = 0.002) and have lower transcutaneous oxygen saturation ([90% (IQR, 90–96), 95% (IQR, 93–96)]; P = 0.032). 76% (38/50) of patients had bilateral parenchymal abnormalities, including 91.30% (21/23) of ICU patients and 62.96% (17/27) of non-ICU patients. 23 Adenovirus pneumonia patients had bacterial infections, 17 had other viruses, and 5 had fungi. Coinfection with virus was more common in non-ICU patients than ICU patients (13 [48.15%]VS 4[17.39%], P = 0.024), while bacteria and fungi not. SMART-COP exhibited the best ICU admission evaluation performance in Adenovirus pneumonia patients (AUC = 0.873, p < 0.001) and distributed similar in coinfections and no coinfections (p = 0.26).

**Funding:** Ming Yan Jiang is supported by the Medical Scientific research project in Xiangtan city (grant number 2020xtyx-6). The funders had no role in study design, data collection and analysis, decision to publish, or preparation of the manuscript.

**Competing interests:** The authors have declared that no competing interests exist.

**Abbreviations:** ICU, intensive care unit; PSI, pneumonia severity index; EBV, Epstein-Barr virus; IQR, interquartile range; AUCs, area under the curves; ROCs, receiver operating characteristic curves; OR, odds ratio; CI, confidence interval; NGS, next-generation sequencing method; PCR, Polymerase Chain Reaction; CT, computed tomography; RSV, respiratory syncytial virus; IFV, influenza virus types A and B; PIV, parainfluenza virus; CMV, cytomegalovirus; CoV, coronavirus; HMPV, human metapneumovirus.

## Conclusions

In summary, Adenovirus pneumonia is not uncommon in immunocompetent adult patients who are susceptible to coinfection with other etiological illnesses. The initial SMART-COP score is still a reliable and valuable predictor of ICU admission in non-immunocompromised adult inpatients with adenovirus pneumonia.

## Introduction

Adenovirus pneumonia, caused by human Adenovirus, consistently reported a high rate of severity and poor prognosis in those with inadequate immunity, such as newborns, hematopoietic stem cell transplant and organ transplant recipients [1, 2]. Recently, it has also been reported frequently in immunocompetent individuals [3, 4]. Heretofore, the etiopathogenesis between non-severe and severe sicknesses in adenovirus pneumonia suffers is still unclear [5]. Early identification of critically ill patients is essential for clinicians [6], which could assist them in conducting proper medical interventions and may improve prognosis. However, few studies have been on early severity recognition in non-immunosuppressed patients of Adenovirus pneumonia. The previous report from immunocompetent adult patients predicted respiratory failure and suggested that initial monocytopenia is a potential predictor [7]. There is little known about the factors that predict admission to the intensive care unit (ICU) in non-immunocompromised Adenovirus pneumonia patients. A recent study from children reported that the severity of Adenovirus infection is significantly correlated with viral load and serotype [8]. Given the difference between children and adults, predicting ICU admission for adenovirus pneumonia in immunocompetent adults is necessary.

Due to the COVID-19 pandemic, some new risk identification tools for viral pneumonia have been reported [9, 10]. Nevertheless, a large retrospective study of COVID-19 showed that recent recognition tools had comparable performance to standard pneumonia prognostic tools [11]. In addition, new tools require more external validation before recommending for their use. More importantly, new tools increase the learning cost of clinicians for learning curve theory. Valuing whether current standard pneumonia scores are appropriately used to predict ICU admission in adenovirus pneumonia can help clinicians quickly identify severe patients using familiar tools.

In this study, we reported the clinical features of Adenovirus pneumonia in immunocompetent adult patients and compared the characteristics between ICU and non-ICU patients. We evaluated current severity prediction tools performance of ICU admission in Adenovirus pneumonia patients to seek an appropriate tool for physicians.

## Patients and methods

### Study design and participants

The research was done at Xiangtan Central Hospital, a tertiary teaching hospital with 2800 beds and 88,694 hospital admissions per year, with approval by Xiangtan Central Hospital Medical Ethics Committee (NO.2020-11-003). Informed consent requirement was waived by Xiangtan Central Hospital Medical Ethics Committee given the study's retrospective nature. From 1 April 2018 to 31 November 2020, all consecutive inpatients were eligible to participate if they presented evidence of Adenovirus infection identified by metagenomic next-generation sequencing method (NGS) or Polymerase Chain Reaction (PCR) test. NGS and PCR test were

standardized performed at BGI Clinical Laboratories (Shenzhen) Co., Ltd and Sansure(Changsha) Biotech Inc, respectively [12].

Patients were not allowed to participate in the trial if they possessed any of the subsequent: a) Immunocompromised; b) No acute lower respiratory symptoms; c) No pneumonia signs on chest radiography or computed tomography (CT); (Fig 1). Immunocompromised patients in CAP were defined as below [13, 14]: (i) Human immunodeficiency virus infection with CD4 + lymphocyte count < 200 cells/μL or percentage < 14% or by the occurrence of an AIDS-defining condition; (ii) hematopoietic stem cell and solid organ transplantation; (iii) receiving chemotherapy prescribed for ≥ 3 months before admission; (iv) receiving corticosteroid use (≥20 mg/day of prednisone for ≥14d or>600mg cumulative dose of prednisone); (v) receiving biological immunotherapy during the last 6 months;(vi) Receiving disease-modify antirheumatic drugs or other immunosuppressive drugs (eg, cyclophosphamide, methotrexate, cyclosporin, hydroxychloroquine); (vii) congenital/genetic immunodeficiencies; (viii)active malignancy or malignancy during 1 year. Given patients with common comorbid conditions such as diabetes, chronic lung disease, or even those who are pregnancy are typically infected with the same spectrum of organisms that cause CAP in healthier adults [13], our study did not exclude these patients.

Patients with Adenovirus pneumonia were defined as: individuals with respiratory symptoms and indications, as well as a new pulmonary infiltrate on chest CT combined with positive detection of Adenovirus. Three experienced pulmonologists independently evaluated all the medical records. The diagnosis of adenovirus pneumonia was discussed, and a consensus was reached.

## Data collection

A team of qualified physicians retrospectively collected the data from electronic medical records, which included demographic, clinical, laboratory, and radiological characteristics, medical interventions, and prognosis. Collecting the underlying condition, signs and symptoms, test results, chest CT scan, and therapeutic measures during hospitalization. Hospitalization stays and cost were collected. Hospitalization expenses include bed fee, medical service fee (doctor's consultation fee, nursing fee), examination fee (CT, B-ultrasound, hematology examination, etc.), drug fee, rehabilitation fee, and the cost of materials used during hospitalization (such as pipes, needles, etc.). Patients were separated into two groups based on their critical care unit hospitalization (Fig 1).

## Image acquisition and analysis

All chest CTs were taken within 24 hours after being admitted to the hospital. When available, CT scans were compared to baseline images to determine the presence of new abnormalities. CT scans were conducted on a 64-slice spiral CT scanner (GE, USA) at 120 kV, automated mAs, and a reconstructed slice thickness of 1.25 mm. Coronal and sagittal reformations were made available. All chest CT images were reviewed retrospectively by two pulmonary radiologists with ten years of expertise. Consensus was established on the final judgments. The Fleischner Society Nomenclature Committee descriptors were used to interpret radiographic CT scans: ground-glass opacities, consolidation, reticular opacities, nodular opacities, and lymphadenopathy. The abnormalities were distributed unilaterally and bilaterally. The presence of an air bronchogram, thickening of the adjacent pleura, and pleural effusions were noted.

## Pathogen detection and lymphocyte subsets analysis

Coinfections were tested in all enrolled patients and determined positive if they met one of the following: 1) detection of virus in respiratory specimens by real-time PCR or NGS, such as

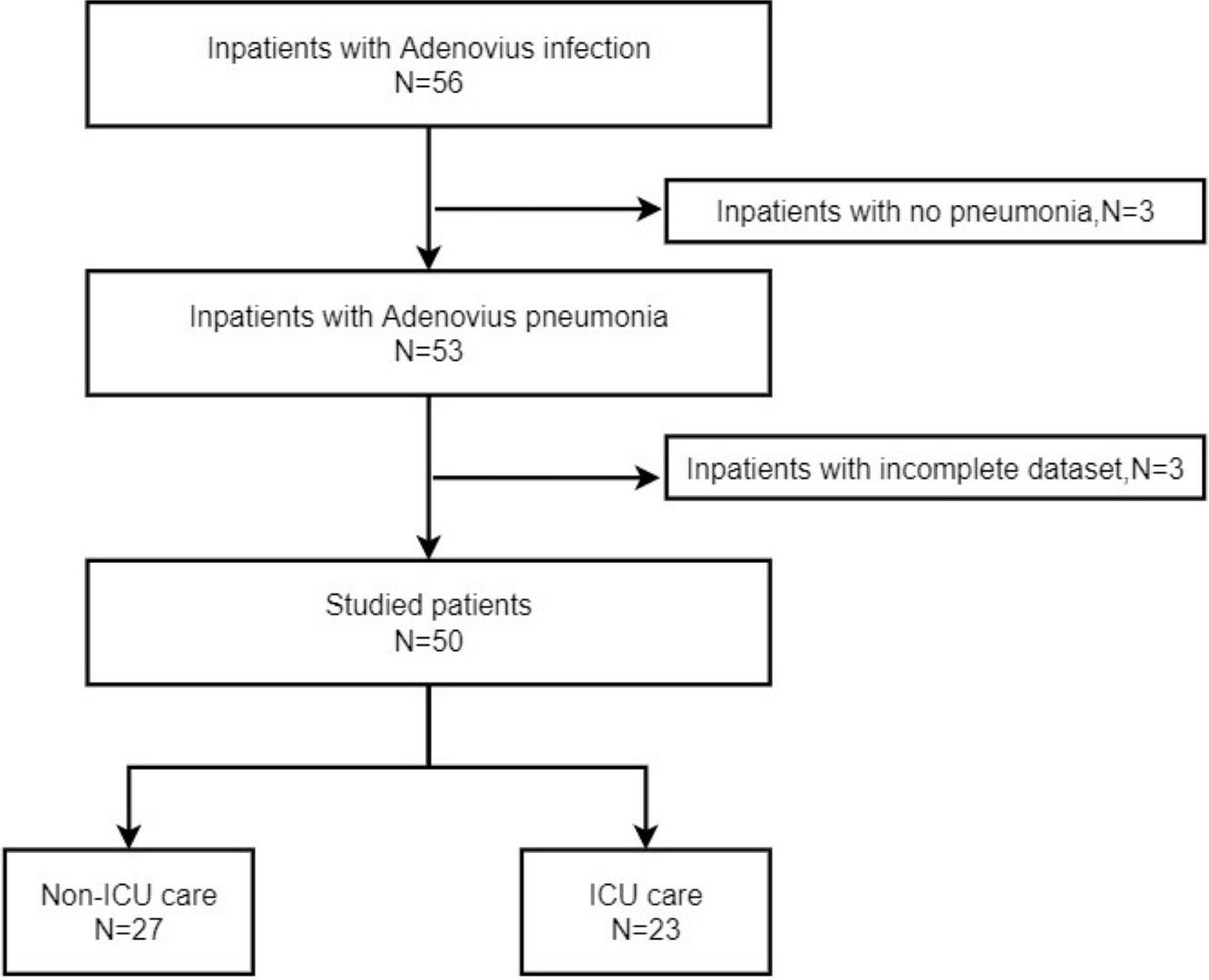

**Fig 1. Enrollment of patients with Adenovirus pneumonia in immunocompetent adults.**

respiratory syncytial virus (RSV), influenza virus types A and B (IFV), parainfluenza virus (PIV), cytomegalovirus (CMV), Rhinovirus, coronavirus (CoV), human metapneumovirus (HMPV), Epstein-Barr virus(EBV); 2) positive respiratory antigen in nasopharyngeal swab for IFV, RSV, ADV and PIV; 3) positive antibody in blood for Legionella pneumoniae, Mycoplasma pneumoniae, Chlamydia pneumoniae, respiratory syncytial virus, Adenovirus, Q fever rickettsia, IFV, and PIV; 4) positive bacterial culture from blood, pleural fluid, qualified lower respiratory tract specimen, qualified urine specimen and so on; 5) detection of bacteria in respiratory specimens by NGS. Peripheral lymphocyte subsets were performed by flow cytometer (BD FACSLyric™, Becton, Dickinson and Company, Inc).

## Severity scores

The severity of Adenovirus pneumonia was evaluated using $PaO_2/FiO_2$ combined lymphocyte, PSI, CURB-65, and SMART-COP. Pneumonia severity scores were calculated on the first day of hospital admission.

## Statistical analysis

SPSS was used to process the information (version 19.0, SPSS Inc., Chicago, IL, USA). Chi-squared was used to compare categorical variables described by frequency rates and percentages. The Mann-Whitney test was utilized to compare median and interquartile range (IQR) values for describing continuous variables. The area under the curves (AUCs) were determined to assess the total prediction accuracy of $PaO_2/FiO_2$, PSI, CURB-65, and SMART-COP for ICU admission using receiver operating characteristic curves (ROCs). The odds ratio (OR) was adjusted with a 95% percent confidence interval (CI). P value < 0.05 was considered statistically significant.

## Results

### Demographic characteristics and clinical features in ICU and Non-ICU patients

In total, 50 hospitalized patients with Adenovirus pneumonia were selected. Of these patients, 27 (54%) were admitted to general wards and 23 (46%) were admitted to or moved to the ICU (Table 1). The mean time to admission to the ICU was 1.04 day with 15 patients admitted directly to the ICU and 5 patients moved later. 86.96% (20/23) of patients who required ICU care were men with a median age of 44 years. Of the 50 patients, 14 (28%) had one or more coexisting medical conditions, including hypertension (6[12%]), cardiovascular disease (5 [10%]), chronic obstructive pulmonary disease (3[6%]), diabetes (3[6%]) and pregnancy (2 [4%]). The main presenting symptoms were fever (47[94%]), cough or sputum (45[90%]), and dyspnea (19[38%]).

Compared with patients who did not receive ICU care (n = 27), patients who required ICU care (n = 23) were more likely to report dyspnea (13[56.52%] vs 6[22.22%]; P = 0.002) and have lower transcutaneous oxygen saturation ($SPO_2$%) [90% (IQR, 90–96), 95% (IQR, 93–96)]; P = 0.032). No differences were observed in temperature, respiratory rate, heart rate, and mean arterial pressure between the two groups. Patients admitted to the ICU had significantly greater platelet counts, aspartate aminotransferase, and D-dimer levels, but lower albumin and globulin concentrations (Table 1). Monocytes was lower in ICU patients than non-ICU patients though no differ significantly. ICU patients had a poorer median oxygenation index than non-ICU patients [236.36 (IQR, 169.70–263.64), 295.23 (IQR, 228.57–342.85)]. 26 patients (52%) used corticosteroid as part of the treatment of pneumonia. Of the 50 patients, 48 patients (96%) had been discharged, whereas two patients (4%) had passed away. Eight of the 23 patients admitted to the ICU required invasive ventilation, and two died both for severe pneumonia. The average time from hospitalization to discharge was ten days (IQR, 8–14.25). Furthermore, the cost of hospitalization in ICU patients was considerably higher than in non-ICU patients (P = 0.003) (Table 1).

### Computed tomographic presentation

Initial chest CTs of 50 enrolled patients were available. The comparison of initial chest CT between the ICU and non-ICU patients was summarized in Table 2. Parenchymal abnormalities were distributed bilaterally in 38 patients, whereas unilateral involvement was shown in 12 patients (Fig 2). Air bronchogram was demonstrated in 33 patients, and pleural effusion was present in 26 patients. About 91.30% (21/23) of patients who required ICU care showed bilateral parenchymal abnormalities, while 62.96% of non-ICU patients showed bilateral parenchymal abnormalities. Among the CT findings, the presence of pleural effusion differed significantly (p = 0.023) between the two groups. Patients who were admitted to ICU showed a

**Table 1. Epidemiological and clinical characteristics of Adenovirus pneumonia patients on admission.**

| Characteristics | All patients (n = 50, %) | ICU care (n = 23, %) | No ICU care (n = 27, %) | P value |
|---|---|---|---|---|
| **Patients** | | | | |
| Age, years | 46(31–56) | 44(36–61) | 49(28–55) | 0.599 |
| Sex | | | | 0.261 |
| Male | 4080.0 | 20(86.96) | 20(74.07) | |
| Female | 10(20.00) | 3(13.04) | 7(25.93) | |
| Smoking | 23(46.00) | 10(43.48) | 13(48.13) | 0.744 |
| **Underlying disease** | 13(26.00) | 6(26.09) | 7(25.93) | 0.783 |
| Hypertension | 6(12.00) | 2(8.70) | 4(14.81) | |
| Cardiovascular disease | 5(10.00) | 4(17.39) | 1(3.70) | |
| Chronic obstructive pulmonary disease | 3(6.00) | 1(4.35) | 2(7.40) | |
| Diabetes | 3(6.00) | 2(8.70) | 1(3.70) | |
| Pregnancy | 2(4.00) | 1(4.35) | 1(3.70) | |
| **Signs and symptoms** | | | | |
| Fever | 47(94.00) | 22(95.65) | 25(92.59) | 0.362 |
| Dyspnoea | 19(38.00) | 13(56.52) | 6(22.22) | 0.002 |
| Myalgia or fatigue | 13(26.00) | 9(39.13) | 4(4.81) | 0.053 |
| Cough or sputum | 45(90.00) | 20(86.96) | 25(92.59) | 0.512 |
| Diarrhoea | 8(16.00) | 5(21.74) | 3(11.11) | 0.312 |
| $SPO_2$, % | 94.5(91,96) | 92(90,96) | 95(93,96) | 0.032 |
| Mean arterial pressure | 90.17(82.33–96.67) | 91.33(84.33–96.67) | 88.33(77.33–97.00) | 0.613 |
| T,°C | 37.85(36.90–38.83) | 38.20(37.30–39.00) | 37.50(36.70–38.10) | 0.106 |
| RR, bpm | 21.00(20.00–22.00) | 22.00(20.00–23.00) | 20.00(20.00–22.00) | 0.113 |
| HR, bpm | 96.00(85.00–110.50) | 96.00(90.00–110.00) | 92.00(85.00–112.00) | 0.459 |
| **Laboratory finding** | | | | |
| White blood cell count, x10$^9$/L | 5.02(4.04–7.62) | 5.05(4.16–7.82) | 4.93(3.84–7.08) | 0.483 |
| Neutrophil count, x10$^9$/L | 4.06(2.58–5.93) | 4.62(3.22–6.44) | 3.66(2.15–5.91) | 0.189 |
| Lymphocyte count, x10$^9$/L | 0.80(0.58–1.10) | 0.80(0.50–0.90) | 0.80(0.60–1.30) | 0.159 |
| Haemoglobin, x10$^9$/L | 131.00(120.75–143.50) | 129.00(120.00–141.00) | 133.00(121.00–145.00) | 0.402 |
| Platelet count, x10$^9$/L | 128.00(98.50–178.25) | 117.00(89.00–171.00) | 141.00(120.00–202.00) | 0.004 |
| Neutrophil To Lymphocyte Ratio, % | 5.45(3.60–8.40) | 6.43(4.26–9.03) | 4.50(2.73–6.69) | 0.044 |
| Lymphocyte To Monocyte Ratio, % | 3.29(2.00–4.18) | 3.33(2.00–4.50) | 2.96(2.00–4.14) | 0.579 |
| Platelet To Lymphocyte Ratio, % | 169.17(124.69–241.78) | 165.45(117.27–253.33) | 170.00(128.18–236.25) | 0.892 |
| Alanine aminotransferase, U/L | 27.85(19.10–48.20) | 38.20(21.30–66.80) | 27.10(16.90–43.60) | 0.119 |
| Aspartate aminotransferase, U/L | 43.00(25.38–93.10) | 69.50(26.20–148.40) | 32.20(22.80–50.00) | 0.006 |
| Albumin, g/L | 35.40(32.05–39.48) | 32.50(28.20–37.60) | 38.50(33.80–40.70) | 0.001 |
| Globulin, g/L | 25.30(23.35–28.40) | 23.90(22.40–27.10) | 26.50(23.90–28.90) | 0.031 |
| Blood urea nitrogen | 3.70(2.99–5.40) | 3.70(2.50–6.50) | 3.60(3.10–4.20) | 0.508 |
| Creatinine, μmol/L | 77.00(63.50–93.50) | 76.00(67.00–93.00) | 78.00(62.00–95.00) | 0.763 |
| Lactate dehydrogenase, U/L | 285.00(198.00–474.50) | 474.50(246.50,741.25) | 217.00(177.00–350.00) | 0.001 |
| Mononucyte count, x10$^9$/L | 0.29(0.17–0.50) | 0.20(0.09–0.39) | 0.32(0.18–0.51) | 0.064 |
| C-reactive protein, mg/L | 72.64(29.44–122.18) | 76.65(35.40–185.71) | 61.06(26.77–112.52) | 0.198 |
| PH | 7.47(7.43–7.50) | 7.46(7.43–7.50) | 7.48(7.43–7.52) | 0.523 |
| Potassium, mmol/L | 132.7(129.08–135.55) | 132.9(130–136) | 132(126–135) | 0.335 |
| Procalcitonin, ug/L | 0.30(0.12–0.92) | 0.21(0.11–0.47) | 0.68(0.13–1.75) | 0.058 |
| D-dimer, mg/L | 0.87(0.46–1.85) | 0.67(0.43–1.36) | 1.29(0.54–3.00) | 0.023 |
| Prothrobin times, s | 13.25(12.68–13.90) | 13.70(12.80–14.20) | 12.90(12.60–13.60) | 0.107 |
| $PaO_2/FiO_2$, mmHg | 244.64(211.77–311.90) | 236.36(169.70–263.64) | 295.23(228.57–342.85) | 0.003 |

*(Continued)*

**Table 1.** (Continued)

| Characteristics | All patients (n = 50, %) | ICU care (n = 23, %) | No ICU care (n = 27, %) | P value |
|---|---|---|---|---|
| Co-infections | 36(72.00) | 17(73.91) | 19(70.37) | 0.783 |
| Use of corticosteroid | 26(52.00) | 17(73.91) | 9(33.33) | 0.003 |
| Tracheal intubation | 8(16.00) | 8(34.78) | 0(0.00) | 0.001 |
| **Prognosis** | | | | |
| In-hospital Mortality | 2(4.00) | 2(8.70) | 0(0.00) | 0.122 |
| Hospital stays, day | 10.00(8.00–14.25) | 12.00(7.00–15.00) | 9.00(8.00–13.00) | 0.249 |
| Hospitalization cost, RMB | 18174.91(13343.24–25051.63) | 22999.69(17502.80–29673.24) | 15002.25(11840.42–20551.88) | 0.003 |

higher probability of pleural effusion. Other CT characteristics, such as ground-glass opacity, lymphadenopathy, air bronchogram, and adjacent pleura thickness, did not change substantially between ICU and non-ICU groups.

## Coinfections pathogen detection and Lymphocyte subsets analysis

The comparison of coinfection results with bacteria, virus, and fungi between the two groups are shown in Table 3. Among the 50 Adenovirus pneumonia patients, 23 had a bacterial infection, 17 had a virus other than Adenovirus infection, and 5 had a fungi infection. The coinfection of virus pathogen was significantly more frequent in non-ICU patients than in ICU patients(P = 0.024). Of the 27 non-ICU patients, 6 (22.22%) patients were coinfected EBV infection, 3(11.11%) were co-infected IFV, 2(7.41%) were coinfected CMV. Likewise, EBV and IFV coinfection were also common in ICU patients. Although bacterial aetiologias were more commonly identified in patients who received ICU care than non-ICU patients, there was no statistical difference(P = 0.054). Regarding bacterial coinfections in ICU patients, *Acinetobacter spp.* (34.78%), *Mycoplasma spp.* (13.04%), and *Mycobacterium tuberculosis* were more frequent. While in non-ICU patients, *Streptococcus spp.* (7.41%), *Klebsiella pneumonia* (7.41%), *Pseudomonas spp.* (13.04%), and *Mycobacterium tuberculosis* were more frequent. Of the coinfected patients, 17 patients had two-or-more organisms infections, and 20 had one. Further, the coinfections of ICU patients were apt to report two-or-more pathogens coinfections than non-ICU patients (p = 0.003).

12 patients admitted to ICU performed peripheral blood subgroup of lymphocyte. Of the 12 ICU patients, 8 had coinfections, and 4 had no coinfections. Absolute numbers of CD3$^-$CD16$^+$CD56$^+$(#), CD3$^-$CD19$^+$(#), CD3$^+$CD19$^-$(#), CD3$^+$CD4$^+$(#) and CD3$^+$CD8$^+$(#) were lower in ICU patients than normal reference value (Fig 3). The CD3$^+$/CD4$^+$ ratio was always within the normal range. The ratios of CD3$^+$CD19$^-$(%), CD3$^+$CD4$^+$(%), CD3$^+$CD8$^+$(%)

**Table 2. Comparisons of initial chest CT findings among Adenovirus pneumonia patients.**

| CT characteristics | All patients (n = 50, %) | ICU care (n = 23, %) | No ICU care (n = 27, %) | P value |
|---|---|---|---|---|
| Lung involvement | | | | 0.021 |
| Unilateral | 12(34.00) | 2(8.70) | 10(37.04) | |
| Bilateral | 38(76.00) | 21(91.30) | 17(62.96) | |
| Ground glass opacity | 3(6.00) | 2(8.70) | 1(3.70) | 0.463 |
| Lymphadenopathy | 6(12.00) | 1(4.35) | 5(18.52) | 0.128 |
| Air bronchogram | 33(66.00) | 16(69.57) | 17(62.96) | 0.627 |
| Thickening of the adjacent pleura | 3(6.00) | 2(8.70) | 1(3.70) | 0.463 |
| Pleural effusion | 26(52.00) | 16(69.57) | 10(37.04) | 0.023 |

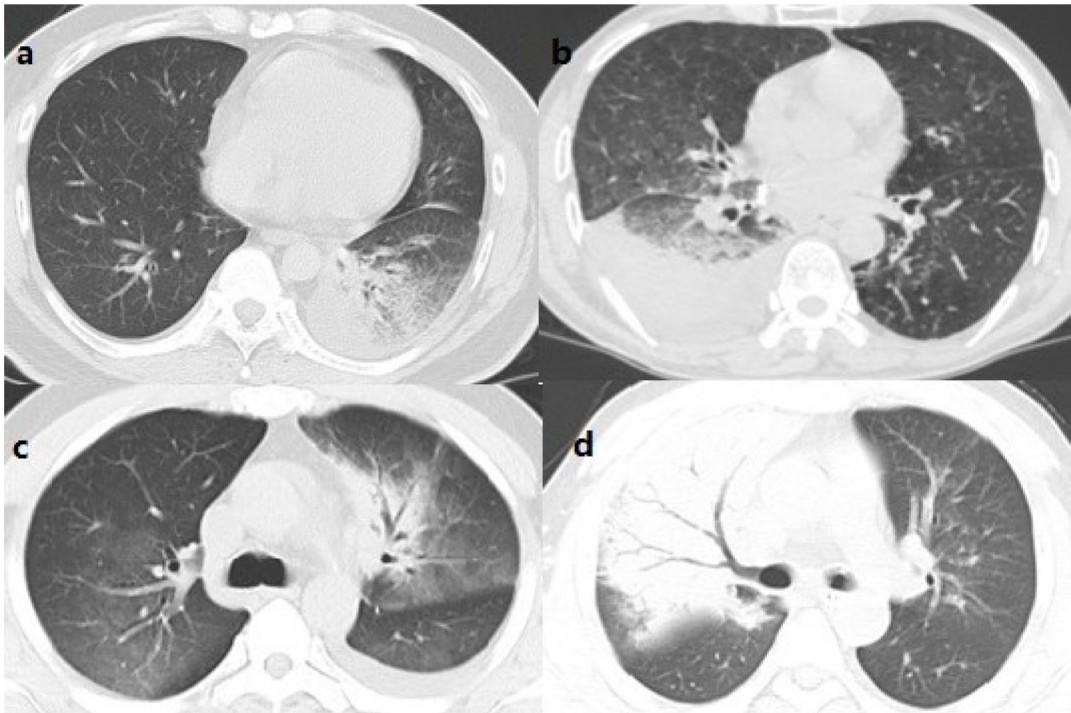

**Fig 2. Chest computed tomography (CT) finding in patients with Adenovirus pneumonia.** (a) Chest CT showed consolidation with interlobular septal thickening in the left lobe. (b) Initial chest CT in a patient who admitted ICU and received noninvasive mechanical ventilation showed consolidation with pleural effusion in the right lobe. (c) Chest CT showed consolidation with ground—glass opacity in left lobe and ground—glass opacity in right lobe. (d) Initial chest CT in a 35 years patient who admitted ICU and received high flow oxygen therapy showed consolidation with air bronchogram in the right lobe.

and CD3$^-$CD16$^+$CD56$^+$(%) were more decreased than normal reference values, while CD3$^-$CD19$^+$ (%) was increased. Compared to non-coinfection ICU patients, Coinfection ICU patients showed a statistically significantly higher decreased rate of CD3$^+$CD19$^-$ (#).

### Predictive scores for severity in Adenovirus pneumonia

To assess the clinical predictive scores for ICU admission, a comparison of ICU patients and non-ICU patients stratified by PSI, CURB-65, and SMART-COP was shown in Table 4. The SMART-COP scores of the two groups were significantly different ($p < 0.001$). Of the 23 patients who required ICU care, 18(78.26%) were identified as a moderate risk, 4(17.39%) as high risk, and 1(4.34%) as very high risk. Moreover, all non-ICU patients were determined to be low or moderate risk (51.85% in low-risk vs. 48.15% in moderate-risk). However, PSI and CURB-65 underrated the severity of ICU admission in a significant number of inpatients with Adenovirus pneumonia ($p = 0.101$; $p = 0.084$, respectively). Of the 23 ICU patients, 17 (74.91%) and 19(82.61%) were stratified to lower risk with I-III of PSI and 0–1 score of CURB-65. Given PaO2/FiO2 combined lymphocyte prediction for severe influenza pneumonia [15], we compared ROCs for ICU admission prediction in patients with Adenovirus pneumonia using four severity scores or indices (PSI, CURB-65, SMART-COP, and PaO$_2$/FiO$_2$) (Fig 4).

Meanwhile, to further explore influence of coinfection on SMART-COP, we divided enrolled patients into two groups (coinfections and no coinfections) and found that the distribution of SAMRT-COP score in two groups was similar (Table 5) with no statistical difference (Fig 5).

**Table 3. The coinfections result of Adenovirus pneumonia patients.**

| Pathogenic types of coinfections | All patients(n = 50, %) | ICU care (n = 23, %) | No ICU care (n = 27, %) | P value |
|---|---|---|---|---|
| **Bacteria** | 23(46.00) | 14(60.87) | 9(33.33) | 0.054 |
| *Acinetobacter spp.* | 9(18.00) | 8(34.78) | 1(3.7) | 0.005 |
| *Pseudomonas spp.* | 2(4.00) | 2(8.70) | 2(7.41) | 0.808 |
| *Klebsiella pneumoniae* | 4(8.00) | 2(8.70) | 2(7.41) | 0.868 |
| *Streptococcus spp.* | 4(8.00) | 2(8.70) | 2(7.41) | 0.868 |
| *Mycoplasma pneumoniae* | 4(8.00) | 3(13.04) | 1(3.7) | 0.230 |
| *Haemophilus influenzae* | 1(2.00) | 0(0.00) | 1(3.70) | 0.356 |
| *Staphylococcus aureus* | 1(2.00) | 1(4.35) | 0(0.00) | 0.279 |
| *Mycobacterium tuberculosis* | 5(10.00) | 3(13.04) | 2(7.41) | 0.512 |
| **Virus** | 17(34.00) | 4(17.39) | 13(48.15) | 0.024 |
| Influenza A virus | 5(10.00) | 2(8.70) | 3(11.11) | 0.779 |
| Influenza B virus | 1(2.00) | 0(0.00) | 1(3.7) | 0.356 |
| Respiratory syncytial virus | 1(2.00) | 0(0.00) | 1(3.7) | 0.356 |
| Epstein-Barr virus | 8(16.00) | 2(8.70) | 6(22.22) | 0.198 |
| Rhinovirus | 1(2.00) | 0(0.00) | 1(3.70) | 0.356 |
| Cytomegalovirus | 2(4.00) | 0(0.00) | 2(7.41) | 0.187 |
| **Fungi** | 5(10.00) | 2(8.70) | 3(11.11) | 0.512 |
| Aspergillus | 2(4.00) | 1(4.35) | 1(3.70) | 0.909 |
| Candida albicans | 3(6.00) | 1(4.35) | 2(7.41) | 0.463 |
| One organism | 20(40.00) | 7(30.43) | 13(48.15) | 0.526 |
| Two or more organisms | 17(34.00) | 10(43.48) | 7(25.93) | 0.003 |

## Discussion

In this study, we described clinical characteristics of Adenovirus pneumonia in immunocompetent adult patients and assessed severity predicting scores for ICU admission. The results shown that initial SMART-COP sores are significantly superior to current CAP (community-acquired pneumonia) severity prediction patterns, such as PSI, CURB-65, and $PaO_2/FiO_2$, for predicting ICU hospitalization in immunocompetent patients with Adenovirus pneumonia. Additionally, we also found that coinfection pathogens distribution differed in ICU patients and non-ICU patients.

SMART-COP score was used to assess the requirement for intensive respiratory or vasopressor support in CAP, which was proposed by Patrick G. P. Charles et al. in 2008 [16]. In the emergency department, SMART-COP score accurately predicted ICU admission with CAP [17]. Unlike other prognosis scores (PSI and CURB-65) explored and aimed to predict 30-day mortality [18], SMART-COP score was mainly explored for clinicians to assess ICU admission severity accurately [19]. Other researchers also verified that risk scores (PSI and CURB-65) were excellent predictors for mortality but not for ICU admission in viral pneumonia [20]. It has been suggested that Adenovirus pneumonia in immunocompetent patients easily occurs in young and previously healthy human [21]. Hence, current severity scores such as PSI and CURB-65 were prone to underestimate the severity of ICU admission.

More importantly, SMART-COP score shows good ICU predictive value in viral pneumonia. In COVID-19, Felippe Lazar Neto et al. investigated 1363 patients and found the SMART-COP score had the best sensitivities for 7-day ICU admission with a negative predictive value beyond 75% [11]. Besides, Bin Cao et al. showed that SMART-COP scores were superior to other scores (CURB-65 and PSI) in predicting influenza virus in ICU admissions [15]. In this study, we demonstrated SMART-COP score is the best accurate prediction of ICU

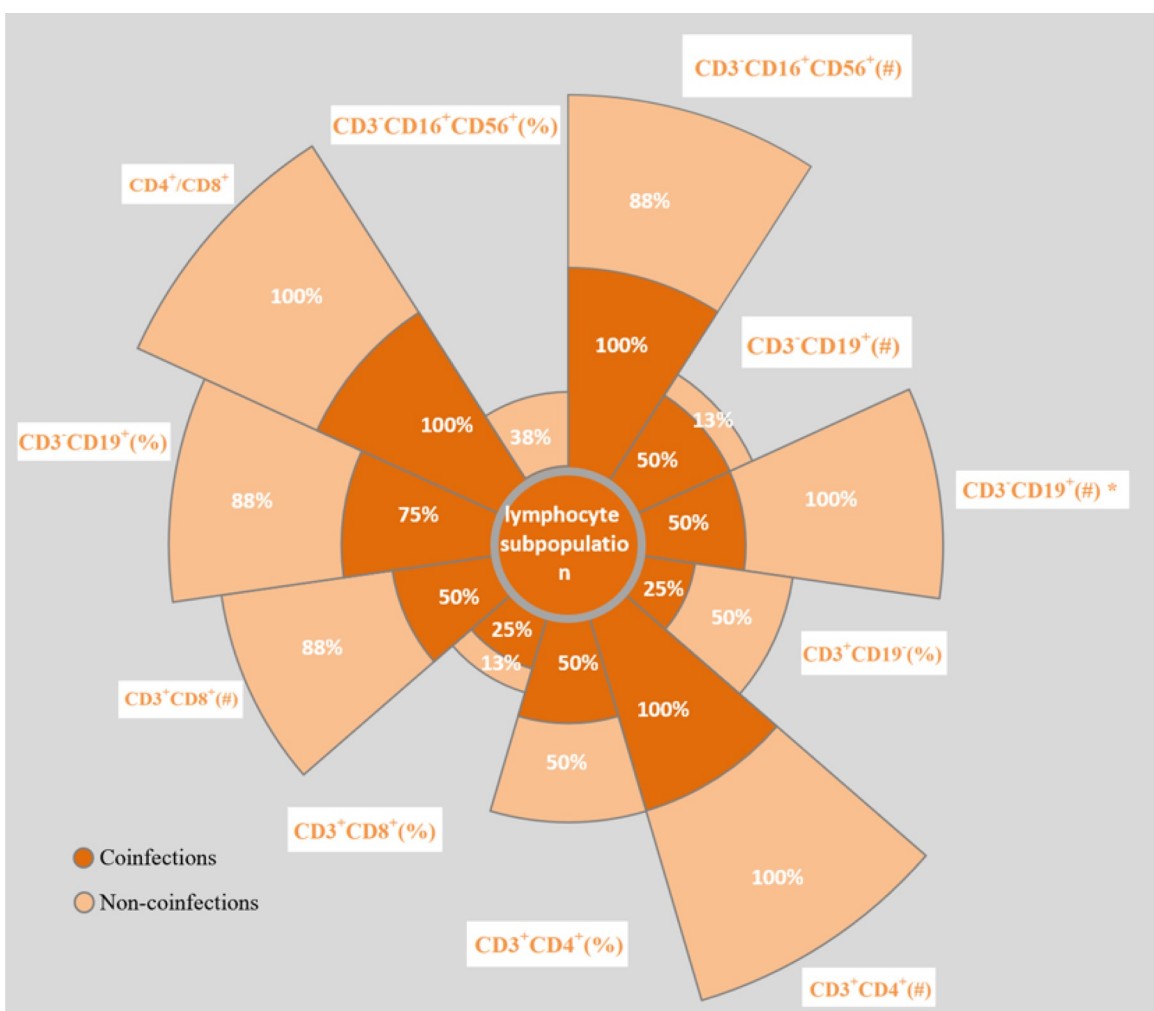

**Fig 3. Comparison of percentage of Peripheral blood lymphocyte subsets in ICU patients with Adenovirus pneumonia according to coinfections.** Percentage in Fig means percentage of patients tested value beyond normal value in all tested patients except CD3⁻CD19⁺ (%) (percentage of lower-than-normal value) and CD3⁺/CD4⁺ (percentage in the range of normal value reference). Normal value references:CD3⁻CD16⁺CD56⁺(#),150-1100M/L; CD3⁻CD19⁺(#),90-560M/L; CD3⁺CD19⁻(#),955-2860M/L; CD3⁺CD4⁺(#), 550-1440M/L; CD3⁺CD8⁺(#),320-1250M/L; CD3⁺CD19⁻(%),56–84%; CD3⁺CD4⁺(%),27–51%; CD3⁺CD8⁺(%),15–44%; CD3⁻CD16⁺ CD56⁺(%),7–40%; CD3⁻CD19⁺(%), 5–18%. * p = 0.046.

admission than other current prediction patterns and the validity of the SMART-COP score was not impacted by coinfections. As a well-known severe score, SMART-COP score has been widely used in clinical practice for decades, which may extend to viral pneumonia in immuno-competent patients.

In additionally, Bin Cao et al. study also confirmed that PaO2/FiO2 combined lymphocyte count was simple and dependent on predicting viral pneumonia ICU admissions [15]. Disap-pointingly, in our study, lymphocyte count combined with PaO2/FiO2 was less predictive than SMART-COP score. One possible reason may be due to the combination pattern of lym-phocyte count and PaO$_2$/FiO$_2$ underestimating ICU admission for lack of variables of chest radiology, which was one of the factors suggestive of progression to ARDS in Adenovirus pneumonia [22].

Coinfection in CAP was usual and had an adverse impact on the severity of the pneumonia [23]. In our study, 72% of patients were coinfected with other pathogens. Previous studies

**Table 4. Comparison of severity pneumonia scores admission in Adenovirus pneumonia patients.**

| Pneumonia Score | All patients (n = 50, %) | ICU care (n = 23, %) | No ICU care (n = 27, %) | P value |
|---|---|---|---|---|
| PSI score | | | | 0.101 |
| Class I | 21(42.00) | 9(39.13) | 13(48.15) | 0.526 |
| Class II | 12(24.00) | 3(14.04) | 9(33.33) | 0.927 |
| Class III | 9(18.00) | 5(21.74) | 4(14.81) | 0.529 |
| Class IV | 7(14.00) | 6(26.09) | 1(3.70) | 0.024 |
| Class V | 0(0.00) | 0(0.00) | 0(0.00) | 1.000 |
| CURB-65 score | | | | 0.084 |
| 0 point | 38(76.00) | 15(65.22) | 23(85.19) | 0.103 |
| 1 point | 10(20.00) | 6(26.09) | 4(14.81) | 0.326 |
| 2 point | 1(2.00) | 1(4.34) | 0(0.00) | 0.279 |
| 3–5 points | 1(2.00) | 1(4.34) | 0(0.00) | 0.279 |
| SMART-COP | | | | < 0.001 |
| Low risk(0-2points) | 14(28.00) | 0(0.00) | 14(51.85) | < 0.001 |
| Moderate risk(3-4points) | 31(62.00) | 18(78.26) | 13(48.15) | 0.030 |
| High risk(5-6points) | 4(8.00) | 4(17.39) | 0(0.00) | 0.025 |
| Very high risk(≥7points) | 1(2.00) | 1(4.34) | 0(0.00) | 0.279 |

showed that patients with Adenovirus pneumonia coinfected with Mycoplasma pneumoniae aggravate the severity of the disease in children [24–26]. According to our research, ICU patients and non-ICU patients had different coinfections pathogen patterns. Compared with those who did not need ICU care, patients requiring ICU admission had a higher coinfections rate of two or more organisms, mostly with bacterial infections. Damage and dysfunction of the epithelial barrier and imbalance of immune function have been suggested as potential causes of viral coinfection with other pathogens [27, 28].

To determine the effects of Adenovirus pneumonia on the body's immune system, patients admitted to ICU performed peripheral blood subgroup of lymphocyte. According to the presence of coinfection, 12 patients were divided into a coinfection group and a non-coinfection group. We analyzed the peripheral blood lymphocyte subsets of critically ill patients and found that the counts of T cells, B cells, and NK cells were all decreased. Accordingly, the levels of $CD3^+$ and $CD4^+$ decreased, and the levels of $CD19^+$ shifted according to the ratio of peripheral blood lymphocytes. We believe that this may be related to the impaired immune function of cells, the impaired function of T cells dependent on B lymphocytes, and the insufficiency of B lymphocytes' antibody production function. Immune responses and hypercytokinemia have been reported to be associated with disease severity in patients with adenovirus infection [29].

Interestingly, coinfected patients requiring intensive care had a lower rate of decline in peripheral blood T-cell counts than non-coinfected intensive care patients. Previous research has demonstrated that a robust inflammatory response can enhance the death and malfunction of T cells, and that T cell depletion is associated with the mortality of Adenovirus-infected individuals [5, 30]. Whether peripheral blood T cells count plays a protective role in coinfections of Adenovirus infection remains to be explored. In brief, hospitalized patients of Adenovirus pneumonia presenting with decreased levels of peripheral blood lymphocytes requires careful assessment of the risks of coinfections.

Several limitations of our investigation should be noted. First, although we tried our best to screen all adenovirus inpatients in our hospital, there were only 50 patients eligible for the study. The conclusion extended to wide use and still needs more large-scale study. Second, as a regional respiratory center with 14 beds and 200 ICU admissions per year, our hospital

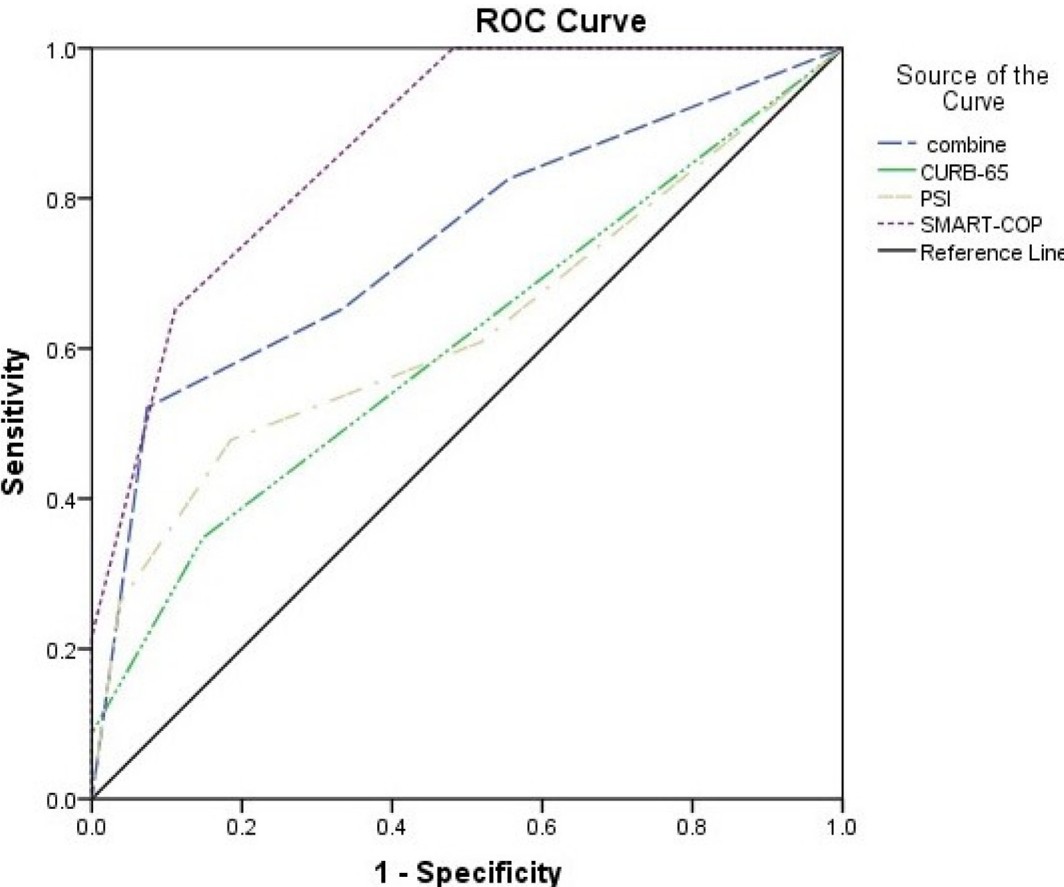

**Fig 4. The ROCs for predicting ICU admission in inpatients with Adenovirus pneumonia.** AUC of SMART-COP was 0.873 (95% confidence interval, 0.779–0.967,p < 0.001);AUC of PaO2/FiO2 combine lymphocyte was 0.742(95% confidence interval, 0.600–0.883, p = 0.004); AUC of PSI was 0.628(95% confidence interval, 0.467–0.789, p = 0.122); AUC of CURB-65 was 0.606 (95% confidence interval, 0.446–0.766, p = 0.199).

undertakes most medical behavior of severe adenovirus pneumonia in Xiangtan, which may cause a selection bias of non-severe patients on account of limited medical resources. Third, our study only assessed lymphocyte subsets in critically ill patients due to a retrospective study,

**Table 5. The distribution of SMART-COP score in coinfections and no coinfections.**

| SMART-COP | ICU care (n = 23) | No ICU care (n = 27) |
|---|---|---|
| Coinfections | | |
| Low risk (0–2 points) | 0 | 12 |
| Moderate risk (3–4 points) | 13 | 7 |
| High risk (5–6 points) | 3 | 0 |
| Very high risk (≥7 points) | 1 | 0 |
| No Coinfections | | |
| Low risk (0–2 points) | 0 | 2 |
| Moderate risk (3–4 points) | 5 | 6 |
| High risk (5–6 points) | 1 | 0 |
| Very high risk (≥7 points) | 0 | 0 |

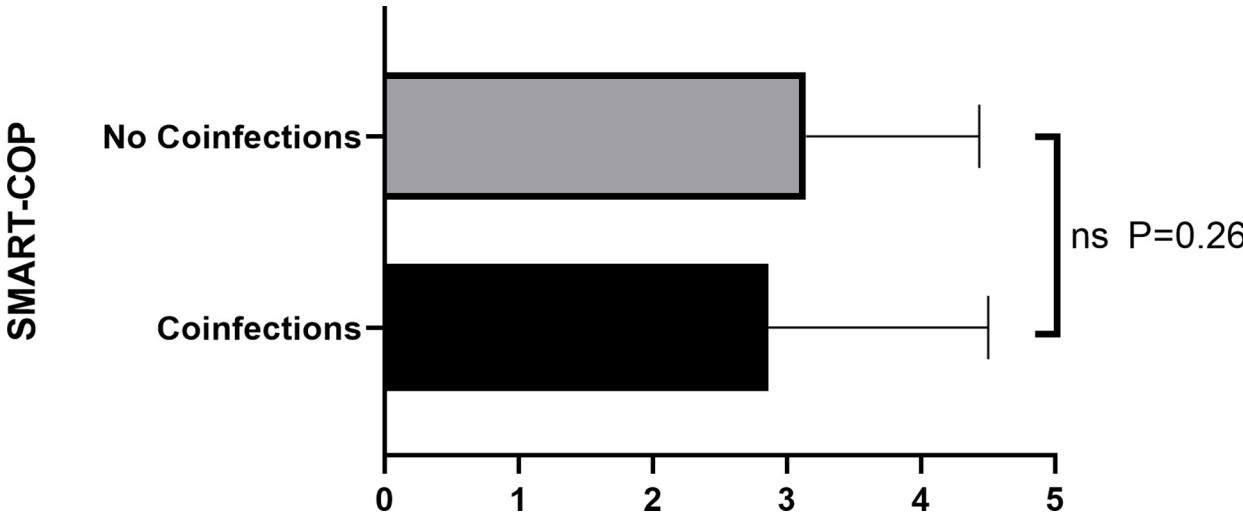

**Fig 5. Comparison of SMART-COP score in coinfections and no coinfections.**

which led to the inability to learn more about the immune function of non-ICU hospitalized patients with Adenovirus pneumonia.

## Conclusions

In conclusion, Adenovirus pneumonia is not rare in immunocompetent adult patients who are prone to coinfect with other etiological infections. Our data suggest that coinfections do not affect the reliability of the SMART-COP score. Tthe initial SMART-COP score is a valid predictive assessment of ICU admission in non-immunocompromised adult hospitalized patients with adenovirus pneumonia.

## Supporting information

**S1 Checklist. STROBE statement—checklist of items that should be included in reports of observational studies.**
(PDF)

## Acknowledgments

All work was performed at the Xiangtan Central Hospital. The authors thanks Qing Wu in the department of Pulmonary and Critical Care for help with data collection. We thank all the patients and physicians involved in the current study.

## Author Contributions

**Conceptualization:** Li Yang, Huan Ming Zhang, Hong Xia, Ming Yan Jiang.

**Data curation:** Chao Hu, Ying Zeng, Li Yang, Hui Li, Huan Ming Zhang, Ming Yan Jiang.

**Formal analysis:** Ying Zeng, Li Yang, Hui Li, Huan Ming Zhang.

**Investigation:** Chao Hu, Ying Zeng, Zhi Zhong, Li Yang, Hui Li.

**Methodology:** Chao Hu, Ying Zeng, Li Yang, Hui Li.

**Project administration:** Hong Xia, Ming Yan Jiang.

**Resources:** Hong Xia, Ming Yan Jiang.

**Software:** Chao Hu, Ying Zeng, Zhi Zhong, Ming Yan Jiang.

**Supervision:** Ying Zeng, Zhi Zhong, Li Yang, Hui Li, Huan Ming Zhang.

**Validation:** Ying Zeng, Zhi Zhong, Li Yang, Huan Ming Zhang.

**Visualization:** Chao Hu, Zhi Zhong.

**Writing – original draft:** Chao Hu, Ying Zeng.

**Writing – review & editing:** Chao Hu, Hui Li, Huan Ming Zhang, Ming Yan Jiang.

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
