## [Decision Letter · Decision Letter 0]

6 Oct 2022

PONE-D-22-17138Clinical characteristics and severity prediction of Adenovirus pneumonia in immunocompetent adultsPLOS ONE

Dear Dr. Jiang,

Thank you for submitting your manuscript to PLOS ONE. After careful consideration, we feel that it has merit but does not fully meet PLOS ONE’s publication criteria as it currently stands. Therefore, we invite you to submit a revised version of the manuscript that addresses the points raised during the review process.

ACADEMIC EDITOR: See comments below.

We look forward to receiving your revised manuscript.

Kind regards,

Muhammad Adrish, MD, MBA, FCCP, FCCM

Academic Editor

PLOS ONE

“This work was supported by the Medical scientific research project in Xiangtan city [grant number 2020xtyx-6]”

4. Please remove your figures from within your manuscript file, leaving only the individual TIFF/EPS image files, uploaded separately.  These will be automatically included in the reviewers’ PDF.

Additional Editor Comments:

Please review comments made by the reviewers and provide point by point response in your revised manuscript. Specifically please address study methodology and criteria used to define immunocompetent adults.

Reviewers' comments:

Reviewer's Responses to Questions

**Comments to the Author**

1. Is the manuscript technically sound, and do the data support the conclusions?

Reviewer #1: Partly

Reviewer #2: Yes

Reviewer #3: Partly

2. Has the statistical analysis been performed appropriately and rigorously? 

Reviewer #1: Yes

Reviewer #2: Yes

Reviewer #3: Yes

3. Have the authors made all data underlying the findings in their manuscript fully available?

Reviewer #1: Yes

Reviewer #2: Yes

Reviewer #3: Yes

4. Is the manuscript presented in an intelligible fashion and written in standard English?

Reviewer #1: Yes

Reviewer #2: Yes

Reviewer #3: Yes

5. Review Comments to the Author

Reviewer #1: thank you for the opportunity to review this manuscript

after reading it there are several doubts and questions that arise.

1) a study period comprising the first year of the covid pandemic was chosen: are the periods before covid and during covid comparable? was the hospital management similar? were the diagnostic studies performed the same? were there no changes that would make the two time periods not comparable?

2)The use of the term "immunocompetent adults" is strongly emphasized, considering that there are diabetics (whose level of metabolic control is unknown) and also patients with chronic obstructive pulmonary disease (COP).

in one of the tables it is mentioned that a significant number of patients have use of sterioids (26 patients), it is not clear if they were used as part of the treatment of pneumonia or were drugs used prior to admission.

3)also in the definition of immunocompetent, was HIV co-infection studied on admission of the patients? no mention is made of whether this was ruled out.

4)The high rate of infection with agents that are not part of the usual community pneumonias is striking. 34% of infection with Acinetobacter spp, also Tuberculosis, even fungal pneumonias (aspergillus which exists almost exclusively in immunocompromised patients). How can this rate of infection be explained?

Reviewer #2: Main points

Abstract

Lines 40-42 Adding all the co-infections (by bacterial, viruses and fungi), could be useful to find differences between ICU and non-ICU patients? It would be interesting to assess.

Methods

Lines 78-81. Include some characteristic of the hospital, such as the total number of hospital admissions per year, number of beds, and ICU.

In line 264 they wrote it is a regional respiratory center, put this also in the methods.

Lines 83-84. Do NGS or PCR are standardized to perform in all adults who are admitted because a pneumonia process? Please clarify.

lines 109-117. Also, searching for other pathogens, antibodies, bacterial cultures, are performed in all patients with low respiratory infections?

Results

Lines 133-134:…23 (46%) were admitted to or moved to the ICU. It should be interesting to know the number of patients who were admitted directly to the ICU, and how many moved later, including the mean time to admission to the ICU.

Lines 144-146. They describe the main results in the laboratory parameters. Should be interesting to comment something about monocytes, because included monocytopenia as a potential predictor in the background (line 59).

Line 148. The two patients who died the cause was directly related with the pneumonia? Another co-infection? septic shock? Please clarify.

Linea 150-151. The cost of hospitalization in ICU were higher. Including costs, and defining how they had calculated, must be included in the methods.

Line 169-170. The abbreviation next to the full name for Epstein-Barr must be first (exchange line 170 with 169). The common abbreviation is EBV, not EB, lines 170 and 277. EBV, should be included in the methods in the part of pathogen detection (lines 109-112)

Line. 170. Do IFA should be IFV? Or what does IFA mean?

Lines 173-175 and table 3. Add the whole gender and specie: Acinetobacter spp, Mycoplasma spp or pneumoniae? Streptococcus spp, Pseudomonas aeruginosa or spp?

Lines 179-187. The searching of lymphocyte subgroups should be added in the methods.

Figure 2 legend is very difficult to read. Describe also, the meaning of the different colors used.

Minor

I suggest the title should establish the severity prediction will be based in a previous score used for pneumonia patients. So maybe could be: “Clinical characteristics and severity prediction score of…….”

Introduction:

Lines 52-53: … with inadequate immunity, such as newborns and organ transplant recipients”. These infections are very common in hematopoietic stem cell transplants. I suggest to add these too.

Methods

line 113. Describe the abbreviations for IFA, or it should be IFV?

Results

Line 134. Approximately 89.96% (20/23) of patients who required ICU…. Approximately? Consider to eliminate the word

Eliminate the two decimals (.00) and leave the whole number

Tables

IN the first column, you can leave the number, example (n=23, %). And eliminate all the symbol % in all the rest of the lines.

Typo

Line 235. Should be pathogens, nor parthenogens

Reviewer #3: Authors evaluated the applicability of severity score in predicting intensive care unit admission of Adenovirus pneumonia such as PSI, CURB-65, SMART-COP, and PaO2/FiO2. The study brought out few important data. The manuscript may be accepted with following few corrections in manuscript:

1. Abstract should have the sample size used for the study.

2. End of the manuscript (line 262) mentioned in there that 56 patients, this needs to be clarified in the methodological section appropriately.

3. Methodological details regarding RT-PCR or NGS should be given - at least some minimal details need to be given.

4. When the manuscript focuses immunocompetent individuals, including some diseased/clinical conditions such as Diabetes and Pregnancy may not be appropriate.

5. Table 3: All the bacteria terms should be initialized.

6. PLOS authors have the option to publish the peer review history of their article (what does this mean?). If published, this will include your full peer review and any attached files.

Reviewer #1: No

Reviewer #2: No

Reviewer #3: **Yes: **Pachamuthu Balakrishnan

---

## [Author Response · Author response to Decision Letter 0]

13 Oct 2022

Response to each point rasied by reviewers are detailed in 'Response to reviewers'file.

---

## [Decision Letter · Decision Letter 1]

16 Nov 2022

PONE-D-22-17138R1Clinical characteristics and severity prediction score of Adenovirus pneumonia in immunocompetent adultsPLOS ONE

Dear Dr. Jiang,

Thank you for submitting your manuscript to PLOS ONE. After careful consideration, we feel that it has merit but does not fully meet PLOS ONE’s publication criteria as it currently stands. Therefore, we invite you to submit a revised version of the manuscript that addresses the points raised during the review process.

We look forward to receiving your revised manuscript.

Kind regards,

Muhammad Adrish, MD, MBA, FCCP, FCCM

Academic Editor

PLOS ONE

Additional Editor Comments:

Please review comments made by the reviewers and provide your response in the revised manuscript.

Reviewers' comments:

Reviewer's Responses to Questions

**Comments to the Author**

1. If the authors have adequately addressed your comments raised in a previous round of review and you feel that this manuscript is now acceptable for publication, you may indicate that here to bypass the “Comments to the Author” section, enter your conflict of interest statement in the “Confidential to Editor” section, and submit your "Accept" recommendation.

Reviewer #1: All comments have been addressed

Reviewer #2: All comments have been addressed

2. Is the manuscript technically sound, and do the data support the conclusions?

Reviewer #1: Partly

Reviewer #2: Yes

3. Has the statistical analysis been performed appropriately and rigorously? 

Reviewer #1: Yes

Reviewer #2: Yes

4. Have the authors made all data underlying the findings in their manuscript fully available?

Reviewer #1: Yes

Reviewer #2: (No Response)

5. Is the manuscript presented in an intelligible fashion and written in standard English?

Reviewer #1: Yes

Reviewer #2: Yes

6. Review Comments to the Author

Reviewer #1: Given the high rate of co-infection with other agents, it is very difficult to assert that a score can predict the severity of adenovirus pneumonia when in reality it is mostly co-infection with multiple agents.

In the conclusions it is stated: "In conclusion, adenovirus pneumonia is not uncommon in immunocompetent adult patients who are prone to coinfection with multiple agents."

This part of the conclusion, in my opinion, if it is possible to extract from the study done.

However, the second part of the conclusion: "Our data suggest that the initial SMART-COP score is a valid predictive assessment of ICU admission in adult non-immunocompromised patients hospitalized with adenovirus pneumonia" does not seem to me to be drawn from the reported study.

Reviewer #2: All the suggestions were made.

It is only important to correct the names of the bacteria: if it has a gender and specie, it is no longer necessary to include spp. (eg. Klebsiella pneumoniae). If it is only the genus without species, must include spp (species), example: Acinetobacter spp. Run lines 204, 205, table 3, and everything that corresponds

7. PLOS authors have the option to publish the peer review history of their article (what does this mean?). If published, this will include your full peer review and any attached files.

Reviewer #1: No

Reviewer #2: No

---

## [Author Response · Author response to Decision Letter 1]

22 Nov 2022

Response to each point raised by reviewers is detailed in 'Response to reviewers' file.

---

## [Decision Letter · Decision Letter 2]

19 Dec 2022

PONE-D-22-17138R2Clinical characteristics and severity prediction score of Adenovirus pneumonia in immunocompetent adultsPLOS ONE

Dear Dr. Jiang,

Thank you for submitting your manuscript to PLOS ONE. After careful consideration, we feel that it has merit but does not fully meet PLOS ONE’s publication criteria as it currently stands. Therefore, we invite you to submit a revised version of the manuscript that addresses the points raised during the review process.

ACADEMIC EDITOR: Reviewer has one final comment. Please address it at your earliest convenience"** in my opinion the separate results for co-infected versus non co-infected should be included and highlight that there was no significant difference, implying that co-infections do not affect the validity of the SMART-COP score."**

We look forward to receiving your revised manuscript.

Kind regards,

Muhammad Adrish, MD, MBA, FCCP, FCCM

Academic Editor

PLOS ONE

Journal Requirements:

Reviewers' comments:

Reviewer's Responses to Questions

**Comments to the Author**

1. If the authors have adequately addressed your comments raised in a previous round of review and you feel that this manuscript is now acceptable for publication, you may indicate that here to bypass the “Comments to the Author” section, enter your conflict of interest statement in the “Confidential to Editor” section, and submit your "Accept" recommendation.

Reviewer #1: All comments have been addressed

Reviewer #2: All comments have been addressed

2. Is the manuscript technically sound, and do the data support the conclusions?

Reviewer #1: Yes

Reviewer #2: Yes

3. Has the statistical analysis been performed appropriately and rigorously? 

Reviewer #1: Yes

Reviewer #2: Yes

4. Have the authors made all data underlying the findings in their manuscript fully available?

Reviewer #1: Yes

Reviewer #2: Yes

5. Is the manuscript presented in an intelligible fashion and written in standard English?

Reviewer #1: Yes

Reviewer #2: Yes

6. Review Comments to the Author

Reviewer #1:** in my opinion the separate results for co-infected versus non co-infected should be included and highlight that there was no significant difference, implying that co-infections do not affect the validity of the SMART-COP score.**

Reviewer #2: The authors made the changes according to the recommendations made in the previous version. I consider it can be published.

7. PLOS authors have the option to publish the peer review history of their article (what does this mean?). If published, this will include your full peer review and any attached files.

Reviewer #1: No

Reviewer #2: No

---

## [Author Response · Author response to Decision Letter 2]

27 Dec 2022

Response to each point raised by reviewers is detailed in 'Response to reviewers' file.

---

## [Decision Letter · Decision Letter 3]

27 Jan 2023

Clinical characteristics and severity prediction score of Adenovirus pneumonia in immunocompetent adults

PONE-D-22-17138R3

Dear Dr. Jiang,

We’re pleased to inform you that your manuscript has been judged scientifically suitable for publication and will be formally accepted for publication once it meets all outstanding technical requirements.

**"Please add  IRB/ethics submission and approval in the methodology"**

Kind regards,

Muhammad Adrish, MD, MBA, FCCP, FCCM

Academic Editor

PLOS ONE

Additional Editor Comments (optional):

Reviewers' comments:

Reviewer's Responses to Questions

**Comments to the Author**

1. If the authors have adequately addressed your comments raised in a previous round of review and you feel that this manuscript is now acceptable for publication, you may indicate that here to bypass the “Comments to the Author” section, enter your conflict of interest statement in the “Confidential to Editor” section, and submit your "Accept" recommendation.

Reviewer #1: All comments have been addressed

2. Is the manuscript technically sound, and do the data support the conclusions?

Reviewer #1: Yes

3. Has the statistical analysis been performed appropriately and rigorously? 

Reviewer #1: Yes

4. Have the authors made all data underlying the findings in their manuscript fully available?

Reviewer #1: Yes

5. Is the manuscript presented in an intelligible fashion and written in standard English?

Reviewer #1: Yes

6. Review Comments to the Author

Reviewer #1: the requested corrections have been made

the only concern is that in the latest version the mention of the ethics committee that evaluated the project and the informed consent waiver was removed,

I am not sure if I did not find the file, but the IRB/ethics submission and approval must appear in the methodology.

7. PLOS authors have the option to publish the peer review history of their article (what does this mean?). If published, this will include your full peer review and any attached files.

Reviewer #1: No

---

## [Editor Report · Acceptance letter]

7 Feb 2023

PONE-D-22-17138R3 

Clinical characteristics and severity prediction score of Adenovirus pneumonia in immunocompetent adult 

Dear Dr. Jiang:

I'm pleased to inform you that your manuscript has been deemed suitable for publication in PLOS ONE. Congratulations! Your manuscript is now with our production department. 

Kind regards, 

on behalf of

Dr. Muhammad Adrish 

Academic Editor

PLOS ONE